# Tumor-Infiltrating CD45RO^+^ Memory Cells Are Associated with Favorable Prognosis in Oral Squamous Cell Carcinoma Patients

**DOI:** 10.3390/cancers15082221

**Published:** 2023-04-10

**Authors:** Nanako Ito, Sachiko Yamasaki, Tomoaki Shintani, Kensaku Matsui, Fumitaka Obayashi, Koichi Koizumi, Ryouji Tani, Souichi Yanamoto, Tetsuji Okamoto

**Affiliations:** 1Department of Oral Oncology, Graduate School of Biomedical and Health Sciences, Hiroshima University, Hiroshima 734-8553, Japan; 2Center of Oral Clinical Examination, Hiroshima University Hospital, Hiroshima 734-8551, Japan; 3School of Medical Sciences, University of East Asia, Shimonoseki 751-8503, Japan

**Keywords:** MHC class I chain-related molecule A, CD45RO, tumor-infiltrating lymphocyte, oral squamous cell carcinoma

## Abstract

**Simple Summary:**

Tumor-infiltrating lymphocytes (TILs) have long been used to predict the prognosis of solid tumors. TIL levels have been reported to be closely associated with the prognosis of patients with squamous cell carcinoma. In contrast, major histocompatibility complex (MHC) class I chain-related molecule A (MICA), which acts as a ligand for NKG2D in natural killer (NK) cells and CD8+ T cells, is related to a higher survival rate in patients with oral squamous cell carcinoma (OSCC). In this study, we investigated the association of OSCC with TILs and MICA in patients with OSCC and its potential as a biomarker.

**Abstract:**

Background: Tumor-infiltrating lymphocytes (TILs) have been used to predict the prognosis of solid tumors. In this study, we investigated which molecules in TILs play a role in the prognosis of patients with oral squamous cell carcinoma (OSCC). Methods: In a retrospective case-control study, we immunohistochemically evaluated the expression of CD3, CD8, CD45RO, Granzyme B, and the major histocompatibility complex class I chain-related molecule A (MICA) of the histocompatibility complex as predictors of prognosis in 33 patients with OSCC. The patients were classified as TILs^High^ or TILs^Low^ according to the number of TILs for each molecule in the central tumor (CT) and invasive margin (IM). Furthermore, MICA expression scores were determined based on the intensity of the staining. Results: CD45RO^+^/TIL in the nonrecurrent group were significantly higher than those in the recurrent group in the CT and IM areas (*p* < 0.05). The disease-free survival/overall survival rate of the CD45RO^+^/TILs^Low^ group in the CT and IM areas and the Granzyme B^+^/TILs^Low^ group in the IM area was significantly lower than that of the CD45RO^+^/TILs^High^ group and the Granzyme B^+^/TILs^High^ group, respectively (*p* < 0.05). Furthermore, the MICA expression score of tumors around the CD45RO^+^/TILs^High^ group was significantly higher than that of the CD45RO^+^/TILs^Low^ group (*p* < 0.05). Conclusions: A high ratio of CD45RO-expressing TILs was associated with a disease-free/overall survival improvement in OSCC patients. Furthermore, the number of TILs that express CD45RO was associated with the expression of MICA in tumors. These results suggest that CD45RO-expressing TILs are useful biomarkers for OSCC.

## 1. Introduction

Tumor-infiltrating lymphocytes (TILs) have long been used to predict the prognosis of solid tumors because they serve as signifiers of the interaction between the immune system and cancer cells [1]. The tumor microenvironment plays an important role in tumor behavior and treatment response. Recently, inflammatory cells have attracted significant attention as a prognostic factor and biomarker. Among the many different inflammatory cell types in the tumor microenvironment, TILs have been extensively studied and have been reported to be closely related to cancer prognosis [2,3]. Furthermore, clinical studies have been conducted in breast, colorectal, and other cancers to examine the relationship between TILs and malignancies, and some pathologists believe that TILs have a higher predictive value than the traditional staging of TNM [4]. Moreover, for head and neck squamous cell carcinoma (HNSCC), TIL levels have been reported to be closely associated with the prognosis of patients with HNSCC. Patients with TIL levels greater than 70% have a better prognosis, so TIL levels may serve as an independent predictor of HNSCC recurrence [4].

The positive correlation between a high number of CD3^+^ and CD8^+^/TILs, and clinical outcomes indicates its potential as a useful biomarker in HNSCC patients treated with definitive chemoradiotherapy [5]. Nguyen et al. reported that higher levels of CD4 ^+^ and CD8^+^/TIL were associated with significantly longer overall survival (OS) and disease-free survival (DFS) in HNSCC patients [6]. Most previous studies have focused on the predictive effects of CD3^+^, CD4^+^, CD8^+^, and CD45RO^+^/TILs on patients’ prognosis [5,7,8,9]. The high number of CD68^+^ macrophages and the expression of CD163 in both macrophages and cancer cells were correlated with poor overall survival (OS) and had a significant impact on the prognosis of OSCC [10]. However, it remains controversial which molecules in TILs better reflect the prognosis in patients with OSCC. Zeromski et al. (1993) showed that cells with CD8^+^ and natural killer (NK) cell phenotypes frequently appeared and were present primarily in the tumor mass of laryngeal SCC [11]. They further reported that the extent of CD8^+^ T cells, especially NK cells, in tumor masses is associated with the expression of major histocompatibility complex (HLA) class I antigens on tumor cells [12,13]. These findings suggest that the presence of HLA class I on tumor cells facilitates the invasion of cytotoxic lymphocytes into the tumor mass [12,13]. According to a study[13], patients with laryngeal cancer whose HLA class I antigens are downregulated as a result of less CD8+ T cell infiltration have a significantly lower survival rate.

MHC class I chain-related molecule A (MICA) is a glycoprotein normally found on the plasma membranes of a small number of human epithelial cells. MICA has been implicated in the pathogenesis of human cancers and may be induced in mucosal epithelial cells and tumor cells exposed to stresses such as chemical substances, ultraviolet light, microbial infection, and carcinogenesis [14,15]. MICA is a ligand for NKG2D, an activating cell surface receptor expressed by NK cells and CD8^+^ T cells, and NKG2D is essential for NK cell activation. In fact, it is believed that NKG2D activation signals in NK cells contribute to the establishment of diverse immune responses [16,17]. In patients with severe tumors, cellular immunity, including NK cell activity, has been reported to be diminished [18]. MICA is expressed on the surface of cancer cells, inducing the activation of the innate immune system, which primarily involves NK cells.

We recently reported that OSCC patients with the MICA A5.1 allele have a higher survival rate than those with other alleles due to NKG2D receptor-mediated NK cell signaling activation [19]. There have been no reports on the relationship between cancer cell MICA expression and TILs to date.

This study’s objective was to assess the prognostic significance of various subpopulations of TILs in OSCC patients. Specifically, we investigated the potential influence of CD3^+^, CD8^+^, CD45RO^+^, and Granzyme B^+^ TILs in OSCC in relation to disease stage and treatment modality. Additionally investigated was the relationship between TILs and MICA expression levels in OSCC.

## 2. Material and Methods

### 2.1. Patients and Specimens

This retrospective case-control study included 33 patients diagnosed with OSCC in the Department of Oral and Maxillofacial Surgery at Hiroshima University Hospital between August 2003 and November 2006. The following are the criteria for inclusion: the availability of archived biopsy samples prior to treatment and clinical data and pathological diagnosis of SCC. Living patients had at least 6 months of follow-up care. Exclusion criteria included a history of surgery on the primary tumor, chemotherapy, and radiotherapy. Before treatment, formalin-fixed, paraffin-embedded (FFPE) tissue samples were obtained. From the date of surgery, the last day of chemoradiotherapy, and the date of photodynamic therapy, DFS or OS was computed (PDT). Data collected from the medical chart of the patients were age, sex, site of lesion, treatment details, and disease classification on the TNM classification of the International Union for Cancer Control (UICC), 6th Ed. 

### 2.2. Immunohistochemical Analysis

Paraffin-embedded tissues were cut at 4–6 μm thickness and immuno-stained to evaluate the expression of CD3, CD8, CD45RO, Granzyme B, and MICA in every OSCC sample [19]. Subsequently, the sections were exposed to protein block 5% normal horse serum (Thermo Fisher Scientific, Waltham, MA, USA) and 2% normal goat serum (Thermo Fisher Scientific) and incubated overnight at 4°C with mouse monoclonal anti-MICA (Nichirei Biosciences, Inc., Tokyo, Japan), mouse monoclonal anti-CD3 (PS1, Nichirei Biosciences, Inc.), mouse monoclonal anti-CD8 (C8/144B, Nichirei Biosciences, Inc.), mouse monoclonal anti-UCHL (UCHL1, Nichirei Biosciences, Inc.) for CD45RO, and mouse monoclonal anti-Granzyme B (GrB-7, Nichirei Biosciences, Inc.), respectively [20,21]. The sections were then exposed for 1 h to a peroxidase-conjugated secondary antibody, and 3,3-diaminobenzidine was used to detect positive staining (DAKO). Slides were counterstained with hematoxylin (Sigma, St. Louis, MO, USA) and mounted in Mount-Quick (Fisher Scientific, Houston, TX, USA) [22,23].

We assessed TILs as lymphocytes in the tumor’s center (CT) and invasive margin (IM) for each sample (Figure 1A). Images were taken with a Nikon Eclipse E800 microscope at 400× magnification (Nikon Instruments Inc., Tokyo, Japan), and the number of TILs in each of the three fields was counted. Samples were sorted in order of cell density: the top 50% of cases were designated as the TILs^High^ group, and the bottom 50% were designated as the TILs^Low^ group in both the CT and IM regions.

Subsequently, using an optical microscope, tumor MICA expression was evaluated. The tumor MICA expression intensity was rated as follows: 0, negative; 1, low; 2, moderate; and 3, high. Then, the total score of positive tumor cells in one field of view (×400 magnification) was calculated, and the intensity of tumor MICA expression was determined as the average of the three fields of view.

### 2.3. Statistical Analysis

JMP 15 statistical software was used for the statistical analysis (SAS Institute Inc., Cary, NC, USA). Next, using the Mann–Whitney U test, the differences between continuous variables were analyzed. Using the Kaplan–Meier method and log-rank test, a survival analysis was conducted. The associations between the examined TIL expressions and DFS/OS rate were evaluated using Cox proportional hazards model-estimated hazard ratios with a 95% confidence interval (CI). Cox regression analyses incorporating an interaction term between CD45RO and Granzyme B expressions and selected patient characteristics (age, tumor stage, and treatment method) were conducted to determine if these variables impeded the effect of TIL expression on DFS/OS rate. *p* < 0.05 was deemed to be significant.

### 2.4. Ethical Considerations

The Human Genome and Genome Research Ethics Committee of Hiroshima University approved this study (approval number: epidemiology—2022-0165) in accordance with the Declaration of Helsinki. The protocol for the study was posted on these websites. Patients could opt out of the study if they were unwilling to provide consent. The requirement for informed consent was waived.

## 3. Results

### 3.1. Clinicopathological Characteristics

Of the 33 patients with OSCC included in this study, 13 were men, and 20 were female (Table 1), while the mean age at the initial examination was 66.2 ± 12.9 years. The primary tumor site varied as follows: tongue cancer, 17 cases; gingival carcinoma of the lower jaw, 10 cases; gingival carcinoma of the upper jaw, 3 cases; carcinoma of the buccal mucosa, 1 case; and plantar carcinoma of the mouth, 2 cases. UICC stage I was 4 cases; stage II was 14 cases; stage III was 3 cases; stage IVA was 10 cases; and stage IVB was 2 cases, respectively. For disease management, surgical resection of the tumor was performed in 48.4% of cases, chemoradiotherapy in 36.4%, and PDT in 15.2%; 22 participants exhibited recurrence, while 11 did not.

### 3.2. Comparison of TILs Density between Recurrence and Nonrecurrence Groups

The immunohistochemical detection of CD3, CD8, CD45RO, and Granzyme B expression was performed (Figure 1B). After treatment, TIL density was compared between the recurrence and nonrecurrence groups. In the nonrecurrence group, median CD45RO^+^/TIL counts (interquartile range [IQR]) were significantly higher in the nonrecurrence group (CT 35.0 [IQR 26.9–38.9] cells/field; IM 115.3 [IQR 91.4–173.2] cells/field) than in the recurrence group (CT 23.5 [IQR 8.9–28.2] cells/field; IM 55.9 [IQR 33.7–116.0] cells/field) (*p* < 0.05; Figure 2). In both the CT and IM regions, the number of other immune cells tended to be higher in the nonrecurrence group compared to the recurrence group.

### 3.3. Comparison of DFS and OS Rates between the TILs^High^ and TILs^Low^ Groups

The disease-free survival (DFS) rate of the CD45RO^+^/TILs^High^ group was significantly greater than that of the CD45RO^+^/TILs^Low^ group in both the CT (low vs. high: mean 64.4 vs. 125.9 months; *p =* 0.0045) and IM (low vs. high: mean 10.4 vs. 80.9 months; *p =* 0.0003) regions (Figure 3). In addition, the DFS rate of the Granzyme B^+^/TILs^High^ group was significantly higher than that of the Granzyme B^+^/TILs^Low^ group on IM (low vs. high: mean 15.9 vs. 73.5 months; *p =* 0.0091) area (Figure 3).

The OS rate of the CD45RO^+^/TILs^High^ group was significantly higher than that of the CD45RO^+^/TILs^Low^ group on both CT (low vs. high: mean 64.4 vs. 125.9 months; *p =* 0.0007) and IM (low vs. high: mean 55.4 vs. 158.7 months; *p =* 0.0031) areas (Figure 4). On IM, the DFS rate of the Granzyme B^+^/TILs^High^ group was also significantly greater than that of the Granzyme B^+^/TILs^Low^ group (low vs. high: mean 75.3 vs. 108.9 months; *p =* 0.0155) (Figure 4). Cox regression analyses incorporating an interaction term between CD45RO^+^ and Granzyme B^+^ expressions and particular patient characteristics were conducted. CD45RO^+^/TILs^High^ in the IM area and Granzyme B^+^/TILs^High^ in the IM area demonstrated a significant correlation for the DFS in multivariate analysis (Table 2). In addition, the hazard ratio of the CD45RO^+^/TILs^High^ group in both the CT and IM areas exhibited a significant correlation with the OS in multivariate (Table 2).

### 3.4. Comparison of Tumor MICA Expression Score between TILs^High^ and TILs^Low^ Groups 

We performed an immunohistochemical analysis on OSCC tissues to examine the expression of MICA in tumor cells and TIL density. Furthermore, MICA was observed in tumor tissues, including the nucleus, membrane, and cytoplasm of cancer cells (Figure 5A,B). Similarly, we found that CD3^+^, CD8^+^, CD45RO^+^, and Granzyme B^+^ TILs were present in both the CT and IM areas. Additionally, we compared the MICA expression score in OSCC between TILs^High^ and TILs^Low^ groups. Tumors with high expression of MICA exhibited significantly increased immune cell densities with CD45RO^+^/TILs^High^ in both CT and IM areas (*p* < 0.05; Figure 5C).

## 4. Discussion

Several studies have demonstrated that a high density of TILs (particularly CD8^+^ T cells) is associated with a more favorable prognosis for many cancer patients [22,23,24]. A positive correlation has also been reported between the increase in CD8^+^/TILs and the prognosis of patients with head and neck cancer [25,26]. Moreover, it has been demonstrated that an increase in CD8^+^/TILs increases chemoradiosensitivity in oropharyngeal or nasopharyngeal cancer [27]. In breast cancer, CD45RO^+^/TILs were associated not only with an antitumor effect but also with the prevention of tumor recurrence [28]. In this study, to determine which molecules expressed by TILs are associated with the prognosis of OSCC patients, we performed a comprehensive immunohistochemical analysis on pre-treatment FFPE tissue sections.

First, a comparison of the number of TILs in the recurrence and nonrecurrence groups revealed that the number of TILs tended to be higher in the nonrecurrence group in both CT and IM areas and that the number of CD45RO^+^/TILs was significantly greater. Several studies have demonstrated that TILs at different locations in the tumor, in CT, or IM areas, have various functions in distinct prognostic prediction and tumorigenesis [5,29]. Our results were almost identical to those of previous reports [5,25]. The impact of four types of TILs on DFS/OS rates was then investigated. Our results showed that both DFS/OS rates in the CD45RO^+^/TILs^High^ were significantly higher than that in the CD45RO^+^/TILs^Low^ in both CT and IM areas. In the IM area, DFS/OS rates in the Granzyme B+/TILs^High^ group were also significantly higher than in the Granzyme B/TILs^Low^ group. Multivariate analysis revealed that CD45RO^+^/TILs and Granzyme B/TILs in the IM area were the independent prognostic factors for DFS/OS in OSCC patients. These results are similar to a recent report which shows that CD45RO^+^/TILs is a significant prognostic factor for DFS but not OS in HNSCC patients [30]. With respect to TIL localization, the CD8^+^/TILs^High^ group has a better prognosis than the CD8^+^/TILs^Low^ group in the IM area; however, there have been no previous reports on CD45RO^+^/TILs [29]. Yajima et al. concluded that CD45RO^+^/TILs may not only help eradicate local tumors, but also prevent metastasis in breast cancer patients [28]. CD45RO^+^/TILs may prevent OSCC recurrence and metastasis based on our findings. However, additional clinical research is required because studies on CD45RO^+^/TILs are scarce [5]. 

MICA is a ligand for NKG2D, an activation receptor for NK cells and certain cytotoxic T cells, and contributes to the establishment of diverse immune responses. In cancer immunity, NKG2D-mediated NK cells attack target cancer cells expressing MICA and comprehensively activate the entire immune system through the production of cytokines such as interferon-γ (IFN-γ) [14,15]. In this study, differences in MICA expression scores between TILs^High^ and TILs^Low^ groups were examined. CD45RO^+^/TILs^High^ showed a significant increase in the MICA expression score, according to the results. We believe that when MICA expression is high in tumor cells, CD45RO^+^ memory cells expressing NKG2D directly exert cytotoxic activity by recognizing MICA.

We recently reported that OSCC patients with the MICA A5.1 allele have a higher survival rate than those with other alleles as a result of the NKG2D receptor-mediated activation of NK cell signaling. Additionally, we demonstrated that the prognosis for patients with the A5.1 homozygous genotype was significantly better than that for patients with the other MICA genotypes [19]. Consequently, genetic polymorphisms of MICA and CD45RO^+^/TILs may serve as potential prognostic factors for OSCC patients. To our knowledge, this is the first study to examine the association between TILs and MICA in OSCC; the results of this study are the first to be reported.

This study has several limitations that must be considered. First, because this is a single-center, retrospective case-controlled study with a small sample size, regional bias may occur. Second, intratumoral heterogeneity may affect the evaluation of TILs and MICA expression, resulting in insufficient detection of immune cells in tumoral tissues. Thirdly, fundamental factors such as smoking history, drinking history, depth, and thickness of cancer invasion were not taken into account.

Nevertheless, this study revealed that the infiltration dynamics of TILs, particularly a high ratio of TILs expressing CD45RO, were associated with DFS/OS improvement in OSCC patients. Additionally, the number of TILs expressing CD45RO was associated with tumor MICA expression. Consequently, our findings suggest that CD45RO^+^/TILs and MICA are useful biomarkers for OSCC.

## 5. Conclusions

This study revealed that the infiltration dynamics of TILs, particularly a high ratio of TILs expressing CD45RO, were associated with DFS/OS improvement in OSCC patients. Additionally, the number of TILs expressing CD45RO was associated with tumor MICA expression. Consequently, our findings suggest that CD45RO^+^/TILs CD45RO-expressing TILs are useful biomarkers for OSCC.

## Figures and Tables

**Figure 1 cancers-15-02221-f001:**
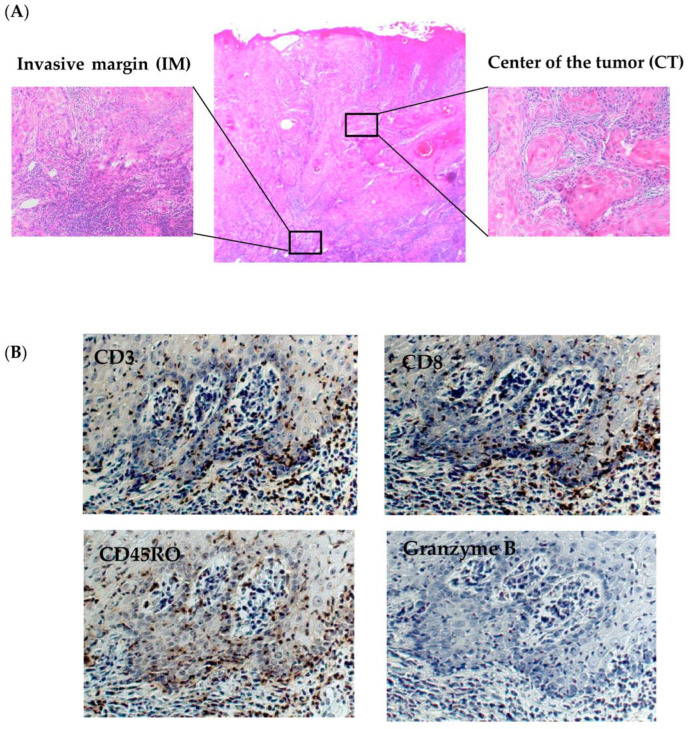
A standardized approach to TIL evaluation for patients with OSCC. The areas of the tumor where the number of TILs was to be measured were determined for CT and IM (**A**) (original magnification center ×40 both sides ×400). Immunohistochemistry was used to identify CD3^+^, CD8^+^, CD45RO^+^, and Granzyme B^+^ TILs (**B**) (original magnification ×400).

**Figure 2 cancers-15-02221-f002:**
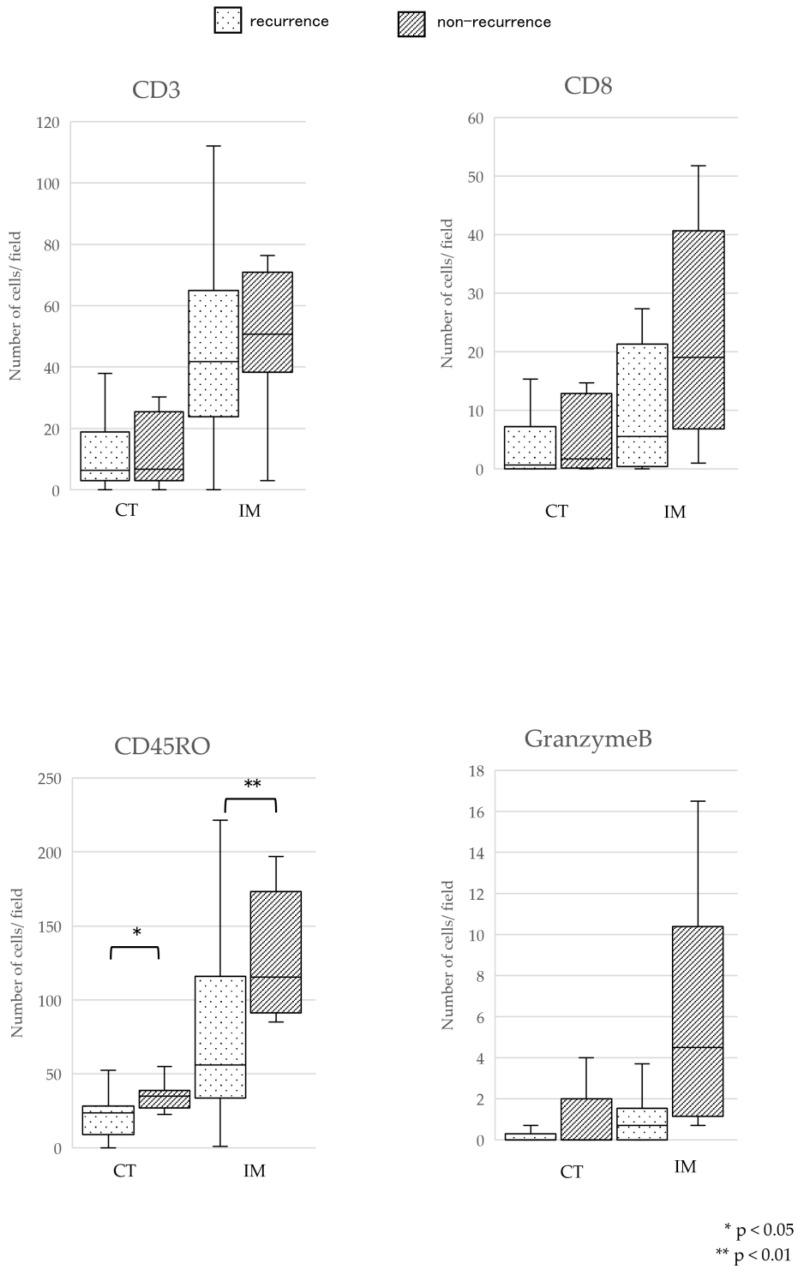
Correlation between recurrence and TIL density in CT and IM. The relationship between TIL density and recurrence was investigated. The number of CD45RO^+^ cells infiltrating was significantly higher in the nonrecurrence group than in the recurrence group in both areas. Differences were determined using the Mann–Whitney U test. * Statistically significant difference at *p* < 0.05. * *p* < 0.05. ** *p* < 0.01.

**Figure 3 cancers-15-02221-f003:**
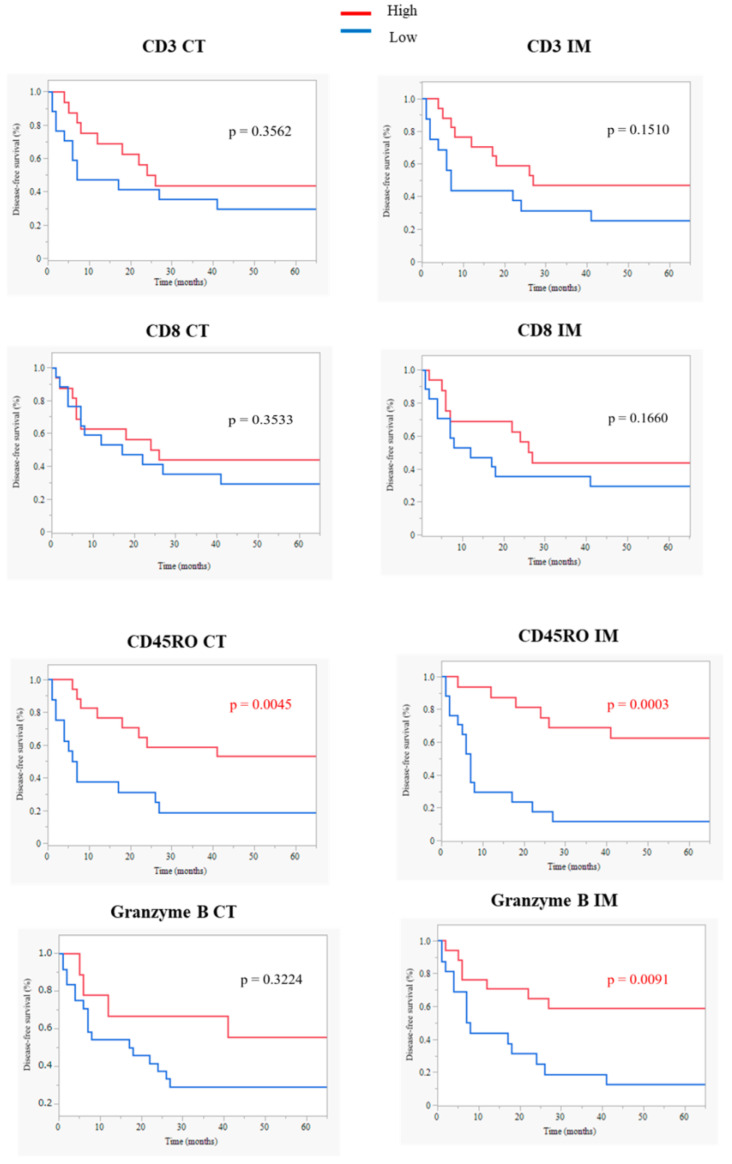
Kaplan–Meier curves of disease-free survival according to the TIL density in OSCC tissue. Each graph showed disease-free survival (DFS) of OSCC patients with high (red line) and low (blue line) numbers of CD3^+^, CD8^+^, CD45RO^+,^ and Granzyme B^+^/TILs in the CT (left panels) or IM (right panels) areas. Kaplan–Meier curves of DFS indicate that patients with CD45RO^+^/TILs^High^ CT (*p* = 0.0045) and IM (*p* = 0.0003) and Granzyme B^+^/TILs^High^ IM (*p* = 0.0091) exhibited significantly improved DFS. Differences were determined using a log-rank test. Statistically significant difference at *p* < 0.05 (red values).

**Figure 4 cancers-15-02221-f004:**
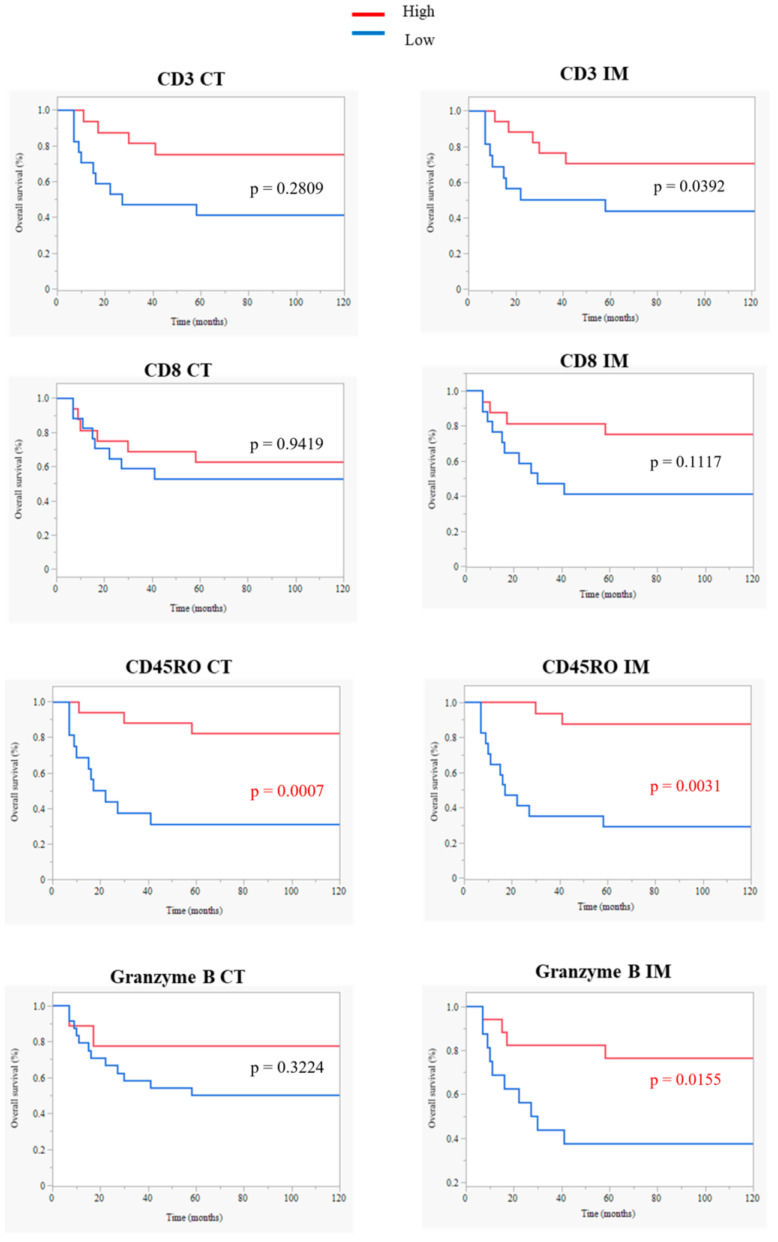
Kaplan–Meier curves of overall survival according to TIL density in OSCC tissue. Each graph showed the overall survival (OS) of OSCC patients with high (red line) and low (blue line) numbers of CD3^+^, CD8^+^, CD45RO^+^, and Granzyme B^+^/TILs in the CT (left panels) or IM (right panels) areas. Patients with CD45RO^+^/TILs^High^ CT (*p* = 0.0007) and IM (*p* = 0.0031) and Granzyme B^+^/TILs^High^ IM (*p* = 0.0155) had a significantly improved OS, as indicated by Kaplan–Meier curves. A log-rank test was utilized to determine differences. Statistically significant difference at *p* < 0.05 (red values).

**Figure 5 cancers-15-02221-f005:**
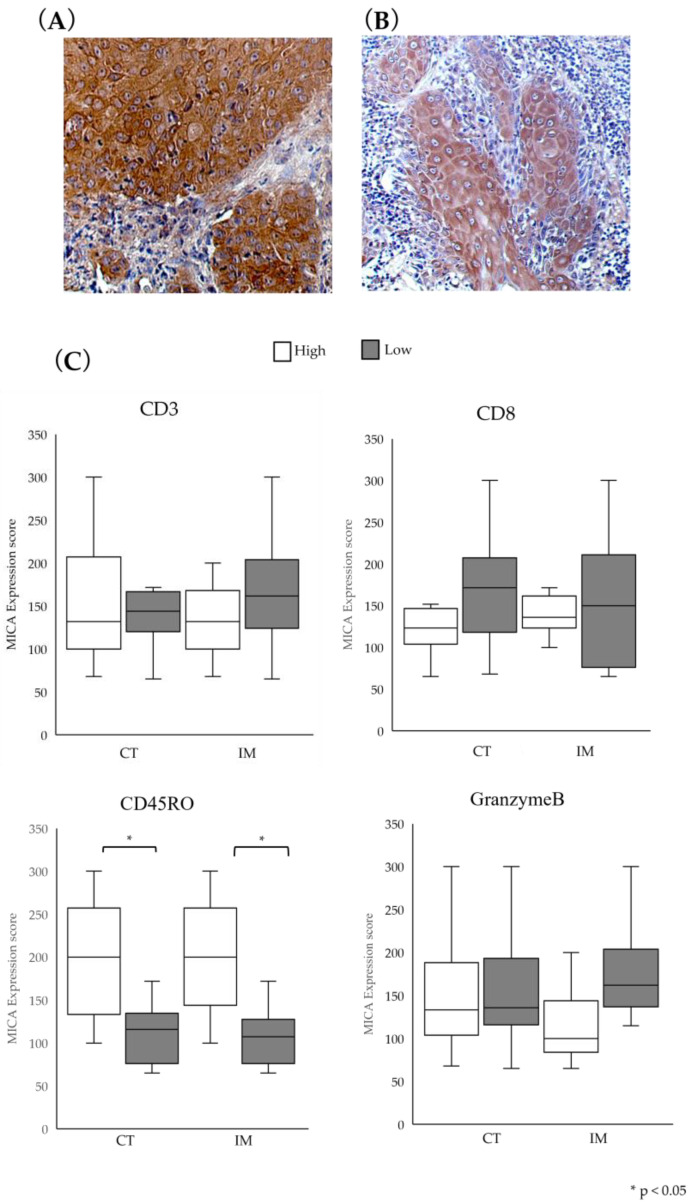
Comparison of the tumor MICA expression score between the TILs^High^ and TILs^Low^ groups. Differences in tumor MICA expression score in the TILs^High^ and TILs^Low^ groups were examined. Whole tumor cells were stained, showing representative examples of low (**A**) and high (**B**) expression (original magnification A and B ×400). The graph indicated the difference in tumor MICA expression scores in the CD3^+^, CD8^+^, CD45RO^+^, and Granzyme B^+^/TILs^High^ and TILs^Low^ groups in the CT and IM regions. In comparison, the tumor MICA expression scores in the CD45RO^+^/TILs^High^ group were more significantly elevated than those in CD45RO^+^/TILs^Low^ group (**C**). Differences were determined using the Mann–Whitney U test. * Statistically significant difference at *p* < 0.05.

**Table 1 cancers-15-02221-t001:** OSCC patients and tumor characteristics.

Characteristics	N	%
OSCC patients	33	
Age (Mean ± SD)	66.2 ± 12.9	
Sex		
Male	13	39.4
Female	20	60.6
Localization		
Tongue	17	51.5
Maxillary gingiva	3	9.1
Mandibular gingiva	10	30.3
Buccal mucosa	1	3.0
Plantar of the mouth	2	6.1
UICC stage		
Ⅰ	4	12.1
Ⅱ	14	42.4
Ⅲ	3	9.1
Ⅳ A	10	30.3
Ⅳ B	2	6.1
Treatment		
Surgery	16	48.4
Chemoradiotherapy	12	36.4
PDT	5	15.2
Recurrence/Nonrecurrence	22/11	66.7/33.3

OSCC oral squamous cell carcinoma; UICC International Union for Cancer Control; PDT photodynamic therapy.

**Table 2 cancers-15-02221-t002:** Results of a multivariate Cox regression analysis for DFS and OS. Cox regression analyses, including an interaction term between CD45RO^+^ and Granzyme B^+^ expressions and patient characteristics, were performed. CD45RO^+^/TILs IM^High^ and Granzyme B^+^/TILs IM^High^ group hazard ratios showed a significant correlation for the DFS in multivariate. The CD45RO^+^/TILs^High^ group’s hazard ratio in both CT and IM exhibited a significant correlation for the OS in multivariate. HR hazard ratio; CI confidence interval; DFS disease-free survival; OS overall survival; CT center of tumor; IM invasive margin.

Parameter (Reference)	DFS			OS		
	HR	95% CI	*p* Value ^a^	HR	95% CI	*p* Value ^a^
Age < 60 years	0.318	0.110–0.918	0.034 *	0.58	0.186–1.814	0.349
Tumor stages 3–4	1.423	0.529–3.832	0.485	1.26	0.436–3.621	0.673
Surgical therapy	0.672	0.231–1.957	0.466	0.316	0.090–1.112	0.073
CD45RO^+^/TILs CT^High^	0.456	0.156–1.334	0.152	0.247	0.062–0.982	0.047 *
CD45RO^+^/TILs IM^High^	0.319	0.112–0.911	0.033 *	0.175	0.047–0.646	0.009 *
Granzyme B^+^/TILs IM^High^	0.385	0.155–0.957	0.039 *	0.687	0.233–2.024	0.496

^a^ Cox proportional hazards model.; * *p* value < 0.05.

## Data Availability

The data presented in this study are available on request from the corresponding author. Publicly available datasets were analyzed in this study.

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
