# Peer review of "Tumor-Infiltrating CD45RO+ Memory Cells Are Associated with Favorable Prognosis in Oral Squamous Cell Carcinoma Patients"

_cancers, 2023, doi:10.3390/cancers15082221_

Round 1
Reviewer 1 Report
The manuscript addressed an important question regarding the immune cell infiltration of oral squamous cell carcinoma (OSCC) and its association with the MICA antigen. The authors demonstrated that a high frequency of CD45RO+ tumor infiltrating lymphocytes is associated with the patients’ survival, which further correlated with the expression of the MICA antigen.
Despite MICA antigen has a key role in NK and T cell-mediated immune responses, the role of MICA has so far not been addressed in OSCC lesions. The report analysed 33 OSCC samples for the expression of T cell markers and MICA and distinguished its expression between the tumor center and margin.
However, although the results are interesting, the patient group is very small to draw any conclusions. In addition, MICA antigen expression should be also analysed in the context of NK cells. Therefore, the second ligand of MICA NKG2AD and NK cells should be analysed.
Furthermore, it should be discriminated whether there are differences in the expression of MICA and patients´ survival regarding the localization of the tumor. Information, whether the patients received a therapy prior to tumor resection and staining would be important. Finally, as minor issue, there are some typos throughout the manuscript, which has to be edited by the authors. This holds also for the figures.
Reviewer 2 Report
The paper presented by Nanako Ito and co-workers introduce a very interesting issue that could have positive repercussions on public health. However paper need to be modified before pubblication.
Abstarct. The conclusion shold be better described it is no clair if CD45RO+ are associated with positive or negative prognosis.
Introduction.
TILS should be better described. line 41-44 are too general.
Line 50-55 the authors should be described the types of cells associated with good or bud prognosis.
Line 66-80 They should introduce the scope of work. It is not clear why the authors chose the tested markers.
Materia and Methods.
Why did the authors select only 33 patients? what is the incidence of this cancer in your country?
Results
line 142- lack of SD
Insert at paraghraph 3.1 the composition of two groups recurrence and non-recurrence
Conclusion.
Line 301-305 the main results should be better resume.
